# Multi-Temperatures Pyrolysis Gas Chromatography: A Rapid Method to Differentiate Microorganisms

**DOI:** 10.3390/microorganisms10122333

**Published:** 2022-11-25

**Authors:** Yun Yang Wan, Ying Jia Zhu, Liang Jiang, Na Luo

**Affiliations:** State Key Laboratory of Petroleum Resources and Prospecting, Research Centre for Geomicrobial Resources and Application, Unconventional Petroleum Research Institute, College of Geosciences, China University of Petroleum, Beijing 102249, China

**Keywords:** microorganism identification, multi-temperatures pyrolysis, thermogravimetric analysis, archaea, bacteria, fungi

## Abstract

The identification of microorganisms using single-temperatures pyrolysis gas chromatography (ST-PyGC) has limitations, for example, the risk of missing characteristic peaks that are essential to the chemotaxonomic interpretation. In this paper, we proposed a new multi-temperature PyGC (MT-PyGC) method as an alternative to ST-PyGC, without sacrificing its speed and quality. Six bacteria (three Gram-positive and three Gram-negative), one micro-fungus and one archaeon, representing microorganisms from different domains, were analyzed by MT-PyGC. It is found that MT pyrograms cover a more complete range of characteristic peaks than ST. Coupling with thermogravimetric analysis, chemotaxonomic information extracted from pyrograms by MT-PyGC have the potential for the differentiation of microorganisms from environments including deep subterranean reservoirs and biomass conversion/biofuel production.

## 1. Introduction

The method for differentiating strains of the same species by pyrolysis chromatography (PyLC) was developed by Reiner, in 1965 [1], and was advanced further to show its feasibility, by the same authors in 1968 [2]. As a new approach to characterizing microorganisms, it has applications in chemotaxonomic analyses of bacteria [3,4,5] and fungi [6] with different single-temperatures (ST). In 2006, Schmidt et al. [7] attempted to optimize the parameters of an ideal ST pyrolysis gas chromatography (ST-PyGC) based on a strain of *Escherichia coli*; however, little research on the topic has been conducted since then. Pyrolysis temperature is the key parameter of PyLC and/or PyGC (MS) [8] that affects the formation of microbes’ volatile components. Numerous studies focus on the identification of microorganisms with single-temperature PyGC (MS). Nonetheless, ST pyrolysis may inevitably produce pyrograms with the same/similar pyrolytic characteristics. However, the optimal ST pyrolysis could vary significantly between samples and between labs.

To the best of our knowledge, there is no universal optimal ST for the identification of more than 20,000 new prokaryotic species using the ST-PyGC method [9]. The identification of bacteria, archaea, and fungi by PyGC at different temperatures (multi-temperatures, MT) in a single run has not been reported. In this paper, a new multi-temperature PyGC (MT-PyGC) method is proposed to simultaneously identify microorganisms in different domains without compromising the speed and quality. Six bacteria (three gram-positive and three gram-negative), one micro-fungus, and one archaeon were chosen and MT-PyGC was run more than four hundred times to develop the methodology, with a focus on normalizing the pyrolysis temperatures for the rapid identification of microorganisms in oil reservoirs, coupled with thermogravimetric analysis (TGA).

## 2. Materials and Methods

### 2.1. Microorganisms and Cultivation

The *E. coli*, *Staphylococcus aureus*, and *Streptococcus bovis* were purchased from the China General Microbiological Culture Collection Center (CGMCC). The *Cupidesulfovibrio oxamicus* was acquired from the National Collections of Industrial, Food and Marine Bacterial (NCIMB). Other microorganisms, including *Acinetobacter calcoaceticus*, *Bacillus subtilis*, *Methanothrix harundinacea*, and *Pycnoporus coccineus*, were isolated in our lab from oil and gas reservoirs [10,11,12] (Table 1).

The microorganisms were cultured, individually, in the corresponding liquid media at optimal conditions [9]. The cultures were examined with microscopy before harvesting by a centrifuge (Eppendorf 5810R, Eppendorf, Hamburg, German) at 4000 r/min for 5 min at 4 °C. The microbes were resuspended in distilled water and centrifuged, and this was repeated three times, to remove residual nutrient broths before lyophilization [13].

**Table 1 microorganisms-10-02333-t001:** Strains for PyGC.

No.	Species	Strain No.	Supplier	Gram ^a^	Domain
1	*Acinetobacter calcoaceticus*	SYO-XJ2LXL20160506	Home lab	G−	Bacteria
2	*Bacillus subtilis*	WD-MHM2014-D3	Home lab	G+	Bacteria
3	*Cupidesulfovibrio oxamicus* [14]	9442	NCIMB	G−	Bacteria
4	*Escherichia coli*	1.1100	CGMCC	G−	Bacteria
5	*Staphylococcus aureus*	1.8721	CGMCC	G+	Bacteria
6	*Streptococcus bovis*	1.1624	CGMCC	G+	Bacteria
7	*Methanothrix harundinacea* [15]	6AC	Home lab	/	Archaea
8	*Pycnoporus coccineus*	WD-MHM2014-ZJ01	Home lab	/	Fungi

^a^ G−: Gram-negative, G+: Gram-positive.

### 2.2. Lyophilization

The cells were transferred to freeze-drying vials (with caps) and were pre-frozen, sequentially, at −20 °C and −80 °C for 30 min and then moved to a vacuum freeze drier (VFD) (SRK GT2-6+, German) for drying. The lyophilization process is shown in Appendix A. The vials were sealed in situ in VFD when the pressure was reduced to below 0.08 mbar. In the initial 10 min, the pressure of the chamber was reduced to 0.48 mbar. After half an hour, the pressure remained at 0.06 mbar. At the same time, the shelf temperature reduced, from room temperature to 3 °C, and the temperature of the freeze-drying chamber remained at −33 °C. The lyophilization process took approximately 2 h. The vials were retrieved and stored in a refrigerator at −20 °C for further analyses [13].

### 2.3. Thermogravimetric Analysis (TGA)

5 mg of the lyophilized microbes were loaded for the TGA analysis with a TG (ZRT-1, with accuracy ± 0.5 °C) [16]. The pyrolysis temperature was programmed, stepwise, to 100, 200, 300, 400, 500, 600, and 800 °C, from room temperature, at a rate of 20 °C per minute. The temperature of 100 °C was maintained for 30 min while the other temperatures remained for 10 min. The data were analyzed using Rsz2000 software.

### 2.4. MT-PyGC

A PyGC system with gas chromatography (PerkinElmer AutoSystem XL, Fitchburg, MA, USA), equipped with an FID detector and a pyrolyzer (SGE Pyrojector-II, Kinesis Australia Pty Ltd., Redland Bay, Qld, Australia), were built to characterize the microorganisms [17]. The microbes (200~500 μg) were pyrolyzed in a quartz tube at temperatures between 300 °C and 800 °C [17]. The pyrolyzates were purged into an Agilent HP-5 column (30 m × 0.25 mm × 0.25 μm, 5% phenyl methyl siloxane) with N_2_ at a pressure of 21 psi (≈144.7 kPa). The inlet temperature was maintained at 250 °C. The flow rate of the carrier gas (N_2_) was 1 mL per minute. The initial temperature of the GC oven was 50 °C. Once the pyrolyzates were purged into the column, the oven temperature remained unchanged for 4 min and was subsequently heated to 200 °C, at a rate of 5 °C per minute, and maintained for 10 min (a running time of 44 min in total) [5]. The data were analyzed with Totalchrom 6.3.1. For quality control, a blank was run after each sample to ensure that the chamber was free of contaminants. The pyroprobe and the inner surface of the pyrolysis chamber were cleaned with acetone before a new strain was analyzed.

## 3. Results

### 3.1. TGA Analysis

The volatile contents in the microorganisms could vary significantly. In Figure 1, eight strains showed different weight loss with a maximum RSD of 6.4% at different temperatures (with accuracy ± 10 °C) after the programmed pyrolysis (Appendix A). The archaeon *M. harundinacea* and sulfate reducing bacteria (SRB) *C. oxamicus* (G−) exhibited a higher temperature resistant characteristic, with a weight loss of less than 50% after pyrolysis from 100 to 800 °C for both the microbes. The anaerobe *C. oxamicus* (G−) lost 36.5% (±0.7%) by 400 °C and only 15.2% (±0.2%) by 800 °C. *E.coli* (G−), *S. bovis* and the fungus *P. coccineus* had similar weight loss in the mid-range, but *S. bovis* showed a different pathway at the low temperatures (100–600 °C). *A. calcoaceticus* (G−), *S. aureus* (G+) and *B. subtilis* (G+) indicated a greater weight loss than the rest of the microbes. Particularly, *B. subtilis* and *A. calcoaceticus* had weight loss of more than 90%.

### 3.2. Pyrograms

The reproducibility of the PyGC was examined. For each strain, at different temperatures (400~800 °C), they were all reproducible. Figure 2 shows the replicates of the pyrolysis of *S. aureus* at 650 °C. The peaks are superimposed with little variation. It should be noted that the fragments in the first 5 min of the chromatogram were a mixture of highly-volatile molecules that were not viable to be separated by the HP-5 column (Figure 2 and Appendix A). Therefore, they are not discussed thereafter in this paper.

The pyrograms of *S. aureus* and *E. coli* at different pyrolysis temperatures are presented in Figure 3. The results showed that *S. aureus* (Figure 3A) and *E. coli* (Figure 3B) had pyrograms with similar retention times, at 500 °C, 600 °C and 800 °C, with the exception of 400 °C. These agreed well with the observations of the TGA analysis (Figure 1). The Fungus *P. coccineus*, bacteria *S. bovis* (G+), *A. calcoaceticus* (G−), and *C. oxamicus* (G−), pyrolyzed at various temperatures, also showed different thermal stabilities, as indicated in the pyrograms (Appendix A). In general, the chromatograms of the right temperatures show more recognizable peaks.

## 4. Discussion

### 4.1. Pyrograms and TGA at Different Temperatures

The weight loss and number of peaks in the pyrograms of *E. coli* and *S. aureus* at various temperatures are plotted in Figure 4. Based on the statistics of the TGA analysis (Figure 1), *S. aureus* (G+) clearly lost more components than *E.coli* (G−) at all pyrolysis temperatures. However, the pyrolysis of *E. coli* resulted in more peaks, e.g., 80 peaks for *E. coli* versus 77 for *S. aureus* at 500 °C and 115 for *E. coli* versus 111 for *S. aureus* at 600 °C. As the pyrolysis temperature increased to 800 °C, the formation of volatile compounds continued; however, the peak number of *E. coli* reduced from 115 to 94, while there was a slight increase for *S. aureus* (113 peaks). This could be ascribed to the stronger heat stability of the cell wall possessed by the gram-positive *S. aureus* belonging to Firmibacteria than that of the gram-negative *E. coli* belonging to Gammaproteobacteria. Interestingly, although the weight loss at 800 °C of *E. coli* (30.08 ± 2.2%) and *S. bovis* (30.62 ± 0.1%) was almost the same, their paths were very different (Figure 3 and Appendix A).

In this study, the volatile components of individual strains at various pyrolysis temperatures were determined by GC. New gas components were generated at different rates. These components, including the microbiomarkers (biomarkers of microorganisms), could be detected by a FID detector (Figure 2, Figure 3 and Figure 4). However, the high pyrolysis temperature, i.e., 800 °C, could further break the volatile compounds down so that these compounds were detectable in low temperatures, but not in high temperatures (Figure 3 and Figure 4). This could explain the phenomena of *S. aureus* displaying more weight loss than *S. bovis* (Figure 1). However, they both had reduced peak numbers at 800 °C (Figure 3 and Appendix A). This finding was quite striking and is in contrast to a previous study [7]. Thus, this suggests that pyrolysis temperatures could greatly impact the quantity of pyrolyzates and their chemical compositions, which contains abundant taxonomic chemical information for distinguishing microorganisms [12].

### 4.2. The Disadvantages of ST-PyGC

ST-PyGC has been documented to differentiate microorganisms, down to the same genus [7] and strains of the same species [18]. In this study, we found that pyrograms of species from different classes using ST-PyGC may not be used to extract chemotaxonomic information. *A. calcoaceticus* (G−), in the Class Gammaproteobacteria, is biochemically different to *S. aureus* (G+) in the Class Bacilli (Firmibacteria). Although there were differences in the TGA results at the same pyrolysis temperature (except for at 650 °C), indicated by their weight loss curves (Figure 1), the two species generated similar components that had close retention times for the characteristic peaks (Figure 5). This was also observed at 800 °C for *S. aureus* (G+) in the Class Actinomycetes and *E. coli* (G−) in the Class Gammaproteobacteria (Figure 3). As the temperature continued to increase, large biomolecules were converted to small molecules that were structurally similar, showing convergence in the pyrograms.

It seems that the fingerprints in the pyrograms for some species may not be used as the signature peaks as they have similar peak characteristics, especially for pyrolysis conducted at 800 °C or higher. As the identification of microorganisms with pyrograms still relies on the recognition of characteristic peaks [1,19], this is one of the drawbacks of ST-PyGC. This is also the reason for the wide range of ST temperatures chosen (400 °C to 900 °C) in the literature, as shown in Table 2. This adds difficulties for comparisons between peer studies.

**Table 2 microorganisms-10-02333-t002:** Reported pyrolysis temperatures for discrimination of microorganisms by PyGC.

No.		Microorganism			Pyrolysis Temperature (°C)	Year	Reference
	Domain	Genera	Species	Strains			
1	Fungi	*Aspergilli*	4 ^a^	12	900	1970	[6]
2	Bacteria	mycobacteria	9 ^b^	9	850	1971	[3]
3	Bacteria	*Vibrio*, *Aeromonas*	3 ^c^	57	800	1973	[20]
4	Bacteria Fungi	*Acholeplasma*, *Citrobacter*, *Micrococcus* *Saccharomyces*, *Rhizopus*	5 ^d^	5	600	1974	[21]
5	Bacteria	*Bacillus*	5 ^e^	5	800	1977	[4]
6	Bacteria	*Salmonella*	10 ^f^	10	900	1978	[22]
7	Bacteria	*Streptococci*	3 ^g^	35	770	1978	[23]
8	Bacteria	*Bacillus*	4 ^h^	32	850	1980	[24]
9	Bacteria	*Bacillus*, *Staphylococcus*, *Pseudomonas*, *Escherichia*, *Legionella*	10 ^i^	10	1000	1992	[25]
10	Bacteria	bacillus	14 ^j^	14	700	1999	[5]
11	Bacteria	*Listeria*	4 ^k^	4	800	2004	[19]
12	Bacteria	*Escherichia*, *Bacillus*, *Micrococcus*	3 ^l^	3	650	2006	[7]
13	Bacteria	*Salmonella*	1 ^m^	1	600	2007	[26]
14	Bacteria	*Bacillus*	3 ^n^	3	530	2009	[27]
15	Bacteria	*Salmonella*	^o^	42	600	2014	[28]
16	Bacteria	*Cupidesulfovibrio*	2 ^p^	2	400, 800	2021	[14]
17	Archaea Bacteria	*Metallosphaera*, *Pyrococcus*, *Halorubrum* *Halomonas*, *Planococcus*, *Shewanella*, *Thermodesulfovibrio*, *Spirulina*, *Chlorella*	9 ^q^	9	650	2022	[8]
18	Archaea, Bacteria, Fungi	See Table 1	8	8	Multi-temperatures	2017	This study

^a^ 3 strains in each of the species *A. parasiticus*, *A. flavus*, *A. tamarii*, and *A. oryzae*; ^b^ *Mycobacterium tuberculosis*, *M. avium*, *M. intracellulare*, *M. kansasii*, *M. fortuitum*, *M. bovis, M. triviale*, *M. terrae*, and *M. marinum*; ^c^ 45 strains of *V. cholerae*, 1 strain of *V. proteus*, 11 strains of *Aeromonas* sp; ^d^ Three bacteria *A. laidlawii*, *C. freundii*, and *M. luteus*, and two fungi *S. cerevisiae*, *R.nigricans*; ^e^ *B. subtilis*, *B. cereus*, *B. firmus*, *B. alvei*, and *B. coagulans*; ^f^ 10 species all newly combined presently; ^g^ 15, 10 and 10 strains of three species *S. mitior*, *S. mutans*, and *S. sanguis*; ^h^ 8 strains in each of the four species *B. subtilis*, *B. pumilus*, *B. licheniformis*, and *B. amyloliquefaciens*; ^i^ *B. anthracis*, *B. cereus*, *B. thuringiensis*, *B. licheniformis*, *B. subtilis*, *S. aureus*, *S. albus*, *P. jkorescens*, *E. coli*, and *L. pneumophila*; ^j^ 9 strains of *Paenibacillus chondroitinus*, 3 strains of *Bacillus thuringiensis*, one strain of *Brevibacillus brevis* and *Alicyclobacillus acidocaldarius*; ^k^ *L. innocua*, *L. ivanovii*, *L. monocytogenes*, and *L. seeligeri*; ^l^ *E. coli*, *M. luteus*, and *B. megaterium*; ^m^ *S. typhi* has been combined as *S. enterica*; ^n^ 2 strains of *B. subtilis* and one strain of *B. megaterium*; ^o^ no species information available; ^p^ *C. liaohensis* and *C. oxamicus*; ^q^ *M. hakonensis*, *P furiosus*, *H. lacusprofundi*, *H. halodenitrificans*, *P. halocryophilus*, *S. frigidimarina*, *T. islandicus*, *Spirulina*, and *Chlorella*.

Studies have shown that even for species from the same genus, e.g., Bacillus [4,5,24] and Salmonella [22,26] (Table 2), the pyrolysis temperatures could be very different. Thus, the chemotaxonomic information extracted from the ST-PyGC results often cannot be referenced. The changes in the pyrograms at different temperatures are unpredictable, especially without qualitative analysis.

Pyrograms generated by ST may have no full coverage of the characteristic peaks that are essential to the discrimination of microbes. For example, M. harundinacea has a characteristic peak designated as “a” in Figure 6, with a retention time of 20.25 min at 400 °C, but has not been detected at any other temperatures. This peak can be used as a biomarker for the strain. Similarly, the elevation of the pyrolysis temperature with M. harundinacea has another peak “b”, at a retention time of 32.71 min (Figure 6), that is not found at other temperatures. Most importantly, the observation is not specific to this archaeon, but was also observed in bacteria (Figure 2) and fungi (Appendix A). Therefore, pyrolysis using ST could have led to the omission of characteristic peaks that are important to the chemotaxonomic interpretation.

### 4.3. Advantages of MT-PyGC

The characterization of microorganisms through MT-PyGC, rather than using the traditional ST-PyGC, overcomes the aforementioned challenges. The multi-temperature pyrograms cover a full range of characteristic peaks. For example, the pyrograms of *A. calcoaceticus* and *S. aureus* were similar, at 650 °C (Figure 5); however, they were seemingly different at 800 °C (Figure 7). *S. aureus* has more peaks detected in the first 22 min than that of *A. calcoaceticus* under the same condition. The chances of spectral convergence in MT-PyGC are lower than in ST-PyGC. Therefore, the accuracy of identification can be improved significantly.

## 5. Conclusions

The current methods for pyrolysis identification of microorganisms using single temperatures (ST-PyGC) has limitations, including the risk of missing characteristic peaks (microbiomarkers) that are essential to the chemotaxonomic interpretation. In this paper, we proposed a method using multi-temperature pyrolysis for the identification of species from different domains of fungi, bacteria, and archaea. The results demonstrate that MT-PyGC improved the accuracy of differentiation in comparison to ST-PyGC. Most importantly, it is possible to compare results from different labs. MT-PyGC provides a new alternative for the identification of massive microorganisms from environments including deep subterranean reservoirs and of biomass conversion/biofuel production in terms of its speed, affordability, and accuracy. It is advantageous because the study can be rapidly conducted, based on the pyrolysis constituents from inter/intra-species or genera at the whole cell level, without any complex pretreatment, preparation, or data analysis.

## Figures and Tables

**Figure 1 microorganisms-10-02333-f001:**
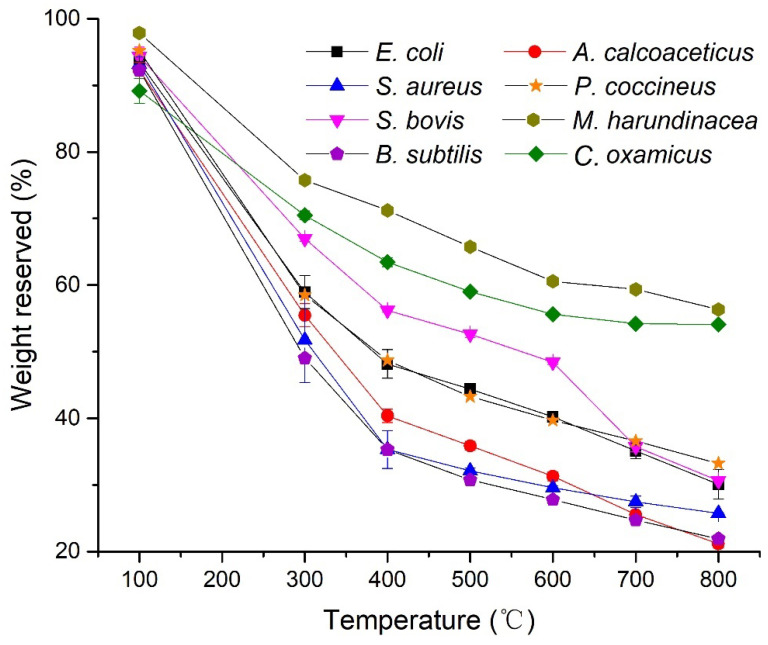
Weight-loss of the eight microbial strains by TGA pyrolysis at different temperatures.

**Figure 2 microorganisms-10-02333-f002:**
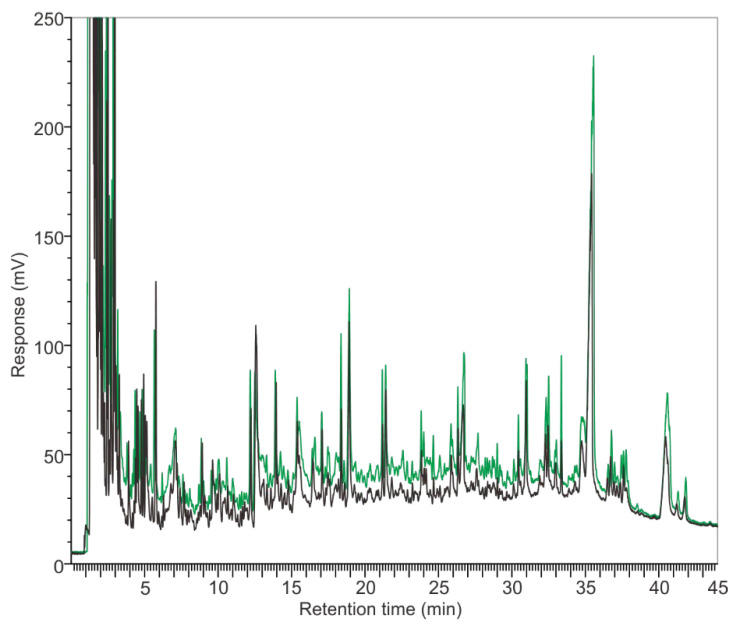
Replication of pyrolysis of *S. aureus* at 650 °C (black and green line).

**Figure 3 microorganisms-10-02333-f003:**
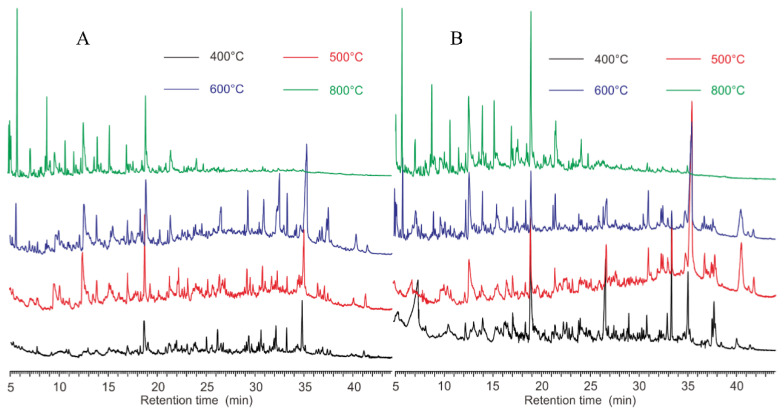
Pyrograms of *S. aureus* (**A**) and *E. coli* (**B**) at four different temperatures.

**Figure 4 microorganisms-10-02333-f004:**
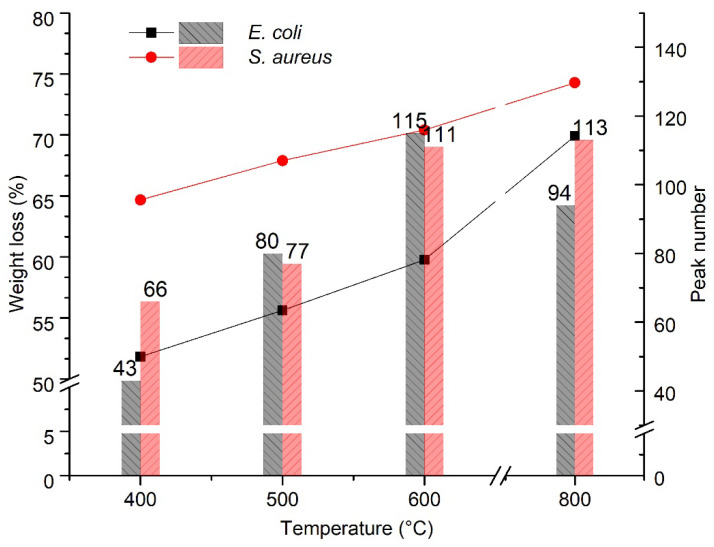
Weight loss and peak numbers of *E. coli* and *S. aureus* at different pyrolysis temperatures.

**Figure 5 microorganisms-10-02333-f005:**
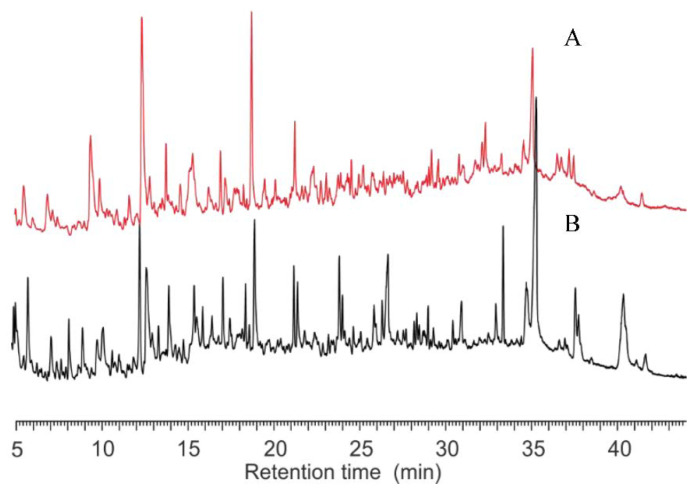
The characteristic peaks of *A. calcoaceticus* ((G−, (A)) and *S. aureus* ((G+, (B)) at 650 °C.

**Figure 6 microorganisms-10-02333-f006:**
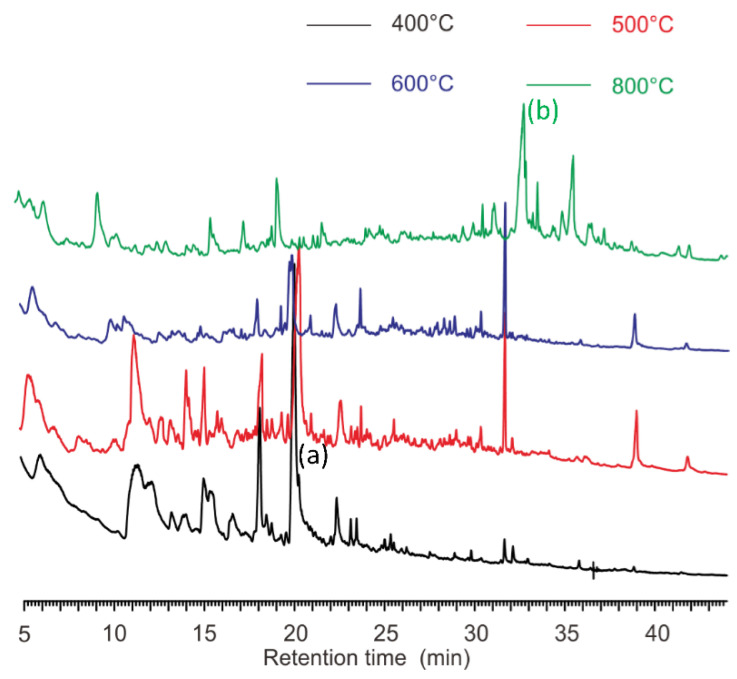
Pyrograms of *M. harundinacea* at four different temperatures. (Peak a): characteristic peak (retention time of 20.25 min) at 400 °C (black line); (Peak b): characteristic peak (retention time of 32.71 min) at 800 °C (green line).

**Figure 7 microorganisms-10-02333-f007:**
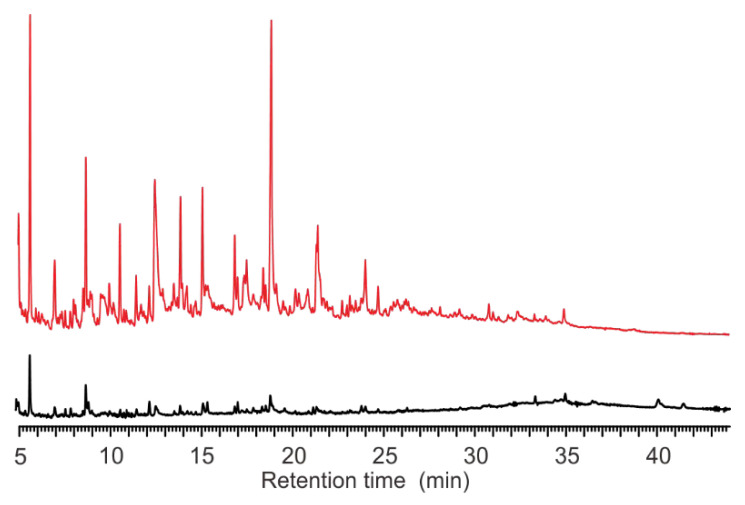
Pyrograms of *A. calcoaceticus* (black line) and *S. aureus* (red line) at 800 °C.

## Data Availability

Not applicable.

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
