# Peer review of "Multi-Temperatures Pyrolysis Gas Chromatography: A Rapid Method to Differentiate Microorganisms"

_microorganisms, 2022, doi:10.3390/microorganisms10122333_

Round 1
Reviewer 1 Report
# Abstract needs a complete overhaul. The scientific language needs to be checked by a native English speaker
# Introduction: The references need to be placed at appropriate positions. eg, reference [1] should come after 1965, (line 65)
#Please indicate a meta-analytical study of the microorganisms discussed for the interest of the readers.
# Please include references in the focused area of the study for the last 5 years
#Improve quality of the figures 2,3,5, 6 and 7. Increase the legend size and resolution. Change Y axis of fig 4 to weight loss.
Author Response
Thanks very much for the review and please see the attachment
# Abstract needs a complete overhaul. The scientific language needs to be checked by a native English speaker
R: Thanks very much. It has been thoroughly modified and polished by native English speakers.
# Introduction: The references need to be placed at appropriate positions. eg, reference [1] should come after 1965, (line 65)
R: Thanks for suggestion and all the references have been put in place.
#Please indicate a meta-analytical study of the microorganisms discussed for the interest of the readers.
R: A good idea but it is out of the scope of this paper and if need we will describe them as a new paper later.
# Please include references in the focused area of the study for the last 5 years
R: The references have been updated thoroughly (partly see Table 2). It shall be pointed out that as we said in text, since 2006, no much more progress in this field has been made. Since the beginning of 21st century, because of its inaccurate and multiple solutions, slow progress was made on PyGC method. More expensive method of Py GC/MS were preferred instead of PyGC. From the cost-effective opinion, we proved MT PyGC instead of ST PyGC is an accurate method and to be one unique solution for species and strains and other rank taxonomy.
#Improve quality of the figures 2,3,5, 6 and 7. Increase the legend size and resolution. Change Y axis of fig 4 to weight loss.
R: Thanks very much and all the figures have been updated with high resolution. Fig 4 has been modified corresponding.

Reviewer 2 Report
A new method to differentiate microorganisms was studied in this work. The new multi-temperatures pyrolysis choromatography (MT-PyGC) procedure is well described and identified here. However, I see more than a new method, a good optimization of the PyGC process varying the temperatures. It would be a good idea to improve the content of the possible new paper with aplication of this new process in some real samples. As I mentioned before, I think this work would better be in an analytical scientific journal, instead of this one. Perhaps, with more experiments besides those made for improving this method, this work could be published in microorganisms.
Author Response
Thank you very much for kind reviews and please see the attachment
A new method to differentiate microorganisms was studied in this work. The new multi-temperatures pyrolysis choromatography (MT-PyGC) procedure is well described and identified here. However, I see more than a new method, a good optimization of the PyGC process varying the temperatures. It would be a good idea to improve the content of the possible new paper with aplication of this new process in some real samples. As I mentioned before, I think this work would better be in an analytical scientific journal, instead of this one. Perhaps, with more experiments besides those made for improving this method, this work could be published in microorganisms.
R: Many thanks for the high evaluation. For the suggestion to be used in real samples, we think it is very good and necessary for the future of the new method. Actually, the new MT PyGC method is itself originated from practical application of rapid identification of oil reservoir microorganisms in field. Ten thousands of reservoir microorganisms preserved in our Petroleum Culture Collection of China were firstly analyzed by MT PyGC/MS. One example, this new method has been documented in the new strains of the new genus Cupidesulfovibrio published in 2021 International Journal of Systematic and Evolutionary Microbiology (Wan et al., Int. J. Syst. Evol. Microbiol. 2021;71:004618). We are making great efforts to promote the application of this new method and hope everyone who can work together with us tightly

Reviewer 3 Report
The manuscript by Wan and coworkers describes multi-temperatures PyGC, as means to differentiate different microorganisms. The idea is good, and perhaps the method can be someday adopted by others. However, the quality of the presentation is low. The introduction is misleading, for instance, it mentions that the method can be used to differentiate strains, however, they are using only strain per specie. Therefore, this method is analyzing different specie no different strains from the same specie, which will be more valuable. Also, across the manuscript, the units are presented using the wrong format (temperature). In summary, this work has some merit but the presentation needs to be improved.
Author Response
Many thanks for the comment and please see the attachment
The manuscript by Wan and coworkers describes multi-temperatures PyGC, as means to differentiate different microorganisms. The idea is good, and perhaps the method can be someday adopted by others. However, the quality of the presentation is low. The introduction is misleading, for instance, it mentions that the method can be used to differentiate strains, however, they are using only strain per specie. Therefore, this method is analyzing different specie no different strains from the same specie, which will be more valuable. Also, across the manuscript, the units are presented using the wrong format (temperature). In summary, this work has some merit but the presentation needs to be improved.
Reply: Thanks very much for the reviewer’s positive comment, encouragement and tentative question.
Because since the first report in 1965 all the previous PyGC literatures (please see the main body of the manuscripts) had all described their works on strains differentiation, we believe the resolution of PyGC for strains of each species is undoubtful, so we then emphasized on the various species. But we also made lots of experiments on various strains of same species, for example strains of A. lowffii (Figure S1) and A. radioresistens (Figure S2).
Figure S1. Strains of Acinetobacter lwoffii from Xinjiang (black line, bottom) and Shengli (red line, top) oilfields, respectively
Figure S2. strains of Acinetobacter radioresistens from Xinjiang (black line, bottom) and Changqing (red line, top)
The temperature symbol and others have been modified.
The language of it has been polished by native English speakers and please see the detail in context.

Round 2
Reviewer 1 Report
The manuscript is acceptable in its present form with minor improvement in the scientific language
Reviewer 2 Report
After reading the new version, the new information of the work shows a clear procedure of identifying microorganisms. Thus, despite the fact that it is a semi analytical work (at the beginning I thought that it would better be in an analytical journal due to the several analytical studies), the system is clearly focused on microorganisms and how identified. In my opinion, this work can be published in this journal.
Reviewer 3 Report
the recommended changes have been addressed.
